# Prevalence and Genetic Characterisation of Human Sapovirus from Children with Diarrhoea in the Rural Areas of Vhembe District, South Africa, 2017–2020

**DOI:** 10.3390/v13030393

**Published:** 2021-03-01

**Authors:** Mpho Magwalivha, Jean-Pierre Kabue Ngandu, Afsatou Ndama Traore, Natasha Potgieter

**Affiliations:** Department of Microbiology, School of Mathematical and Natural Sciences, University of Venda, Thohoyandou 0950, South Africa; Kabue.Ngandu@univen.ac.za (J.-P.K.N.); Afsatou.Traore@univen.ac.za (A.N.T.); natasha.potgieter@univen.ac.za (N.P.)

**Keywords:** hospitalized patients, outpatients, rural communities, sapovirus

## Abstract

Diarrhoeal disease is considered an important cause of morbidity and mortality in developing areas, and a large contributor to the burden of disease in children younger than five years of age. This study investigated the prevalence and genogroups of human sapovirus (SV) in children ≤5 years of age in rural communities of Vhembe district, South Africa. Between 2017 and 2020, a total of 284 stool samples were collected from children suffering with diarrhoea (*n* = 228) and from children without diarrhoea (*n* = 56). RNA extraction using Boom extraction method, and screening for SV using real-time PCR were done in the lab. Positive samples were subjected to conventional RT-PCR targeting the capsid fragment. Positive sample isolates were genotyped using Sanger sequencing. Overall SV were detected in 14.1% (40/284) of the stool samples (16.7% (38/228) of diarrhoeal and 3.6% (2/56) of non-diarrhoeal samples). Significant correlation between SV positive cases and water sources was noted. Genogroup-I was identified as the most prevalent strain comprising 81.3% (13/16), followed by SV-GII 12.5% (2/16) and SV-GIV 6.2% (1/16). This study provides valuable data on prevalence of SV amongst outpatients in rural and underdeveloped communities, and highlights the necessity for further monitoring of SV circulating strains as potential emerging strains.

## 1. Introduction

Diarrhoeal diseases are recognized as the third leading cause of death among children under five years of age in South Africa (SA) [1,2]. The effects of poor sanitation and hygiene practices, quality of supplied water may play an important role in the burden of diarrhoeal disease which is a major concern in developing countries [1,3,4]. Viral infections may present from asymptomatic to relatively mild diarrhoea with a headache and fever, to severe watery diarrhoea accompanied with abdominal cramps [5].

Sapovirus is an enteric virus, and is recognized as a public health problem causing acute gastroenteritis in people of all age groups globally, and it also causes outbreaks in semi-closed settings, like orphanages and elderly care facilities [6]. Sapovirus has been associated with persistence vomiting suggested to possibly cause gastroenteritis in humans [7]. The increase of acute gastroenteritis associated with sapovirus (SV) has been reported and recognized as a major public health problem particularly in developing countries [8,9]. It is documented that after the successful deployment of the Rotavirus vaccine, SVs have emerged as the second most commonly etiological virus behind Norovirus in children with acute diarrhoea [9]. In addition, a longitudinal study by MAL-ED reported SV as a notable second highest attributable incidence of diarrhoea within the enrolled rural community in South Africa [10].

Sapovirus is a single-stranded, positive-sense RNA virus, with three open reading frames (ORFs) identified as ORF1, ORF2, and ORF3, of which ORF1 region is labelled to encode among other proteins, a major capsid protein (VP1) [11,12]. Sapovirus display a high level of diversity, currently with four genogroups (e.g., GI, GII, GVI, and GV) associated with human gastroenteritis infection [6,13,14,15]. Viral particles spread from person to person through faecal–oral route by consuming contaminated food and drinking water, and/or handling sapovirus-positive faeces [16,17,18,19,20].

Prevalence of Sapovirus varies in different countries possibly due to environmental conditions and hygiene practices, which are likely to play a role in the infection frequency of individuals in different settings [21]. Only a few studies in SA have reported on the prevalence of SV from hospitalized patients [3,22,23,24], and a longitudinal investigation [10]. This study aims to report on human SV circulating among children of less than five years of age residing in the rural settings of the Vhembe region in Limpopo, SA.

## 2. Results

### 2.1. Study Population

Out of 284 participants enrolled, 68% (193/284) were outpatients and 32% (91/284) were hospitalized. Amongst the participants, children less than 12 months age group, both in various settings, namely: symptomatic outpatients (66/137: 48.2%), asymptomatic outpatients (38/56: 67.9%), and hospitalized (46/91: 50.5%) were the most enrolled, with the least enrolled age groups from 13 to 60 months in all settings (Table 1).

### 2.2. Sapovirus Detection

The RIDA^®^GENE Sapovirus kit (R-Biopharm AG, Darmstadt, Germany) showed evidence of proficiency for all tested samples. The Ct value of the reactions ranged from 14.10 to 40.43 (mean = 30.89) and was determined from a threshold of 0.03.

Of the 284 collected samples, 40 (14.1%) were positive for human SV by RIDA^®^GENE test kit, with most samples detected at a low viral concentration (C_t_ range of 34.12–42.22). Among these positive samples, 13.2% (12/91) were of the hospitalized cases and 14.5% (28/193) from outpatient cases, with insignificant statistical difference (*p* = 0.765, Pearson Chi-Square, 2-sided). Out of 28 outpatients, 2 (7.1%) were patients without diarrhoea, and 26 (92.9%) were patients with diarrhoea.

### 2.3. Clinical Manifestation

Table 2 shows the manifestation of symptoms stated. In positive cases, diarrhoea as a single symptom was observed in 57.9% (22/38) and also in 42.1% (16/38) with other symptom(s), amongst which vomiting was specified in 29.7% (11/38) cases, fever in 21.6% (8/38) cases, abdominal pains in 10.8% (4/38) cases, and dehydration in 8.1% (3/38) cases. Overall reporting to the health facilities within three days of symptoms manifestation was noted in 63.2% (144/228) cases, while 36.4% (83/228) reported after three days.

Most cases of other symptoms accompanying diarrhoea manifested in multiple (data not shown). Table 3 present the clinical manifestation of symptoms on patients reporting to clinics and those admitted in hospitals. Diarrhoeal symptom was mostly noted in clinic settings as previously emphasized (methods, study population). Of all case patients, only 11.1% of dehydration was recorded in Hospital settings.

### 2.4. Household Setting and SV Distribution

Data on household setting as presented on Table 4, was recorded during an interview prior to sample collection. Notable number of SV positive cases were likely associated with the use of latrine and water sources used, and further correlated by Bayesian linear regression. The correlation between the positive cases and water source showed a statistical significance (*p* = 0.006), whereas there was no significance in the correlation between positive cases and usage of latrine (*p* = 0.067). Children breastfeeding were also observed to be the most infected by the virus but this was not statistical significant (*p* = 0.930). The distribution of SV, was high among children <12 months of age by 47.5% (19/40), followed by 35% (14/40) detection from children aged 13 to 24 months, with the least detection of 10% (4/40) from children aged 25 to 36 months, and 7.5% (3/40) in children 37–48 months of age (Table 4).

As presented in Figure 1, seasonal distribution of SV in this study differed between the sampling periods. Sapovirus was commonly detected in summer seasons during all these years. A high detection rate was noticed during winter in 2017, autumn season in 2018, and spring season in 2019.

### 2.5. Molecular Characterization

Further molecular analysis on the identified SV positive cases was done to determine the SV genogroups, and 16 (40%) of the identified positives were successfully amplified for genogrouping. With note, nine of these samples had a low RNA concentration (C_t_ > 34.12). SV-GI (13/16: 81.3%) was predominately detected followed by SV-GII (2/16: 12.5%) and SV-GIV (1/16: 6.2%). Furthermore, three randomly selected samples (SV-G1-R/SaV124F amplicons of samples number Z01 (C_t_ = 23.42), Z22 (C_t_ = 22.21), and Z31 (C_t_ = 32.85); from different clinics) were subjected to sequencing analysis to determine the SV genotypes. A BLAST search gave a 95–99% sequence identity to the most closely related human SV strains in GenBank (accession number: MT741940, MT741941, and MT741942). Noronet genotyping tool [25] confirmed these sequences as the following SV genotypes: one as G1.1, and two G1.5.

The identified SV genotypes from this study, SVG1.1 (MT741940) showed a 98% identity by clustering with strain detected from a chimpanzee in Congo (KJ858686.1), and 96% identity with strains detected from human stool samples in South Africa (KP196476.1; KP196437.1). The identified SV genotypes from this study SV-G1.5 strains (MT741941 and MT741942) showed 98% identity to a human strain reported in India (KU317439.1) and 96% identity to the strain detected from food (ruditape) in Japan (AB765970.1), which also clustered closely. Other strains which gave identity of between 90% and 95% on GenBank, showed distinct clusters from the strains detected in this study when rooted by a porcine SV strain (DI203382.1), as presented in Figure 2.

## 3. Discussion

The occurrence of SV in outpatients and hospitalised settings has been reported with different rates in various developing countries [3]. In this study, the prevalence rate (14.1%) is high compared to a 7.7% SV detection rate previously reported in SA from hospitalised patients [27], 5.2% [28], and 2.5% [29] reported in Brazil. However comparable detection rate (13.9%) was observed in hospitalized patients from Pakistan [30]. Furthermore, a study in Burkina Faso reported a 9% detection rate from hospitalized and outpatients [31]. A longitudinal investigation by MAL-ED found a high SV detection rate of 22.8% (range of 18.9–27.5%) from around the world, including South Africa [9].

This cross-sectional study report human SV prevalence rate of 14.1% within the rural communities of South Africa, with no evident outbreaks throughout the period of study. SV were detected in both asymptomatic (5%; 2/40) and symptomatic (95%; 38/40) children. A manifestation of more than one symptoms per individual(s) was noted, although it could be as a result of other factors or causative pathogens. As shown in Table 3, most cases of diarrhoea were recorded in the clinics as compared to hospitals. It is assumed that most patients report to the clinics and possibly drink oral rehydration solution for self-treatment, which is effective and less costly [32]. Hence, cases of multiple symptoms including dehydration were seen in hospitals as severe cases.

Evidence of SV associated with diarrhoea amongst children less than five years of age has been observed around the globe [10,33], with viral gastroenteritis frequently seen in infants less than one year of age [5]. This study demonstrate a 17.1% (31/181) of SV detection in children less than two years of age presenting with diarrhoea (Table 1), comparable to a study reported in Peru which documented a 12.4% (37/299) SV detection rate among similar cohorts [9]. The findings of high proportion of SV detection amongst children <1 year of age is of concern in rural settings because of limited health infrastructure, and that children are more vulnerable due to development of their immune [5].

The variability of SV’s prevalence in different regions of the world is evident, although SV-GI is the genogroup that is most associated with severe gastroenteritis cases [34]. In this study, SV-G1 predominance among the patients presenting with diarrhoea was noticeable and highlighted a possible threat among children in the rural communities. The detected SV-G1.1 strain showed identity with those detected from the hospitalized patients elsewhere within South Africa (Figure 2: KP196476.1 and KP196437.1). In this study, a high rate of SV detection in outpatient children compared to the hospitalised children was observed, as previously reported in Nicaragua [35]. This could imply that SVs circulating within rural communities are more likely associated with moderate diarrhoea which does not require hospitalisation. However, Table 2 presents a notable number of patients reporting late to the health care facilities which might lead to limited data documented. There is globally a noted concern of high prevalence rates of SV in low-income countries, but little data on the frequency of human SV in developing countries have been documented to date [3].

Among other provinces in SA, Limpopo is predominantly rural and one of the poorest provinces, with scarce water resources and sanitation [36]. From data collected in this study, the results showed that most households used pit-latrines and municipal tap water as source of water (Table 4). Positive cases associated with water sources (variables: municipal tap water, borehole, river, and spring) accessible by people for drinking, food preparation, bathing and other daily household tasks was established with statistical significance. Poor microbial quality on piped-water (tap water) in a low socio-economic setting, and high level of indicator micro-organism counts in water storage containers compared to the indoor tap water have been previously reported [1,37]. The use of pit-latrines might potentially play a role in viral transmission, directly or indirectly through flies as vectors, as previously stated [38]. Positive cases linked with children breastfeeding in this study is of concern, since risk factors such as poor hygiene practice and overcrowding of people living in a household leading to close contact of persons have been identified as some of the factors associated with diarrhoeal disease [1,39]. In this study occurrence of SV varied throughout the seasons; however, there was an evidence of persistent SV’s manifestation in summer season, which is considered a rainy season in the Vhembe region of South Africa. Furthermore, many rural dwellers use untreated water for domestic use and are more at risk of the devastating effects of diarrhoea since causative pathogens may be transmitted as a result of poor quality of water, inadequate sanitation, and hygiene [1,40].

## 4. Materials and Methods

### 4.1. Study Population

This is a cross-sectional study, conducted between 2017 and 2020. The participants were children ≤5 years of age (with and without diarrhoea) residing in rural communities of Vhembe district in Limpopo Province, South Africa. In South Africa, most cases of intestinal gastroenteritis are seen by the primary health care centres (clinics) situated in the rural communities and only the severe cases (with dehydration) are referred to the hospitals [41].

### 4.2. Sample Collection

To exclude the chances of nosocomial infections from hospitalized participants, samples were collected within the first two days of admission. Additionally, only patients admitted due to the diarrhoeal case were considered for this study. The World Health Organisation [42] definition for diarrhoea was used to include patients in the study. A total of 284 of stool samples (228 diarrhoeal and 56 non-diarrhoeal) were collected from participants at their respective local primary health care centres (20 clinics and 4 hospitals). Availability of samples was dependent on the willingness of participants to provide a sample and be included in the study. The samples collected were kept in closed stool bottles at +4 °C, and transported to the laboratory.

### 4.3. Quality Control

All sample analysis protocol and storage were done in separate rooms to avoid cross-contamination and PCR inhibition. Internal control was used to monitor inhibition and contaminations. For RIDA^®^GENE test runs, C_t_ values for internal control and positive control were in range as following the Quality Assurance Certificate.

### 4.4. Molecular Detection and Genogrouping of Sapovirus

#### 4.4.1. Nucleic Acid Extraction

Prior to the nucleic acid extraction by Boom extraction reagents (Severn Biotech, Worcestershire, UK), the stool specimens were diluted to 10% suspension in Phosphate-Buffered Saline solution (pH = 7.0, Lasec SA (Pty) Ltd., Cape Town, South Africa) and stored at −20 °C. The viral RNA was extracted using the boom extraction method [43], with internal control added during extraction for quality control [41]. Briefly: A 500 µL of 10% suspension stool was centrifuged for 15 s at 13.3 × 1000 g. Then, 900 µL of L6 buffer was added to the supernatant in a sterile 1.5 mL tube, mixed by vortex for 1 min, 20 µL of internal control was added, centrifuged for 15 s at 13,300× *g*. Into a sterile 1.5 mL tube, 100 µL of Silica beads (Severn Biotech, Worcestershire, UK) was added to the transferred supernatant, mixed by vortex for 15 s and shaken softly for 15 min. Tube was centrifuged at 300× *g* for 15 s, and supernatant discarded. The pellet was re-suspended in 500 µL of L2 buffer, centrifuged at 300× *g* for 15 s and supernatant discarded. The pellet was re-suspended in 500 µL of 70% Ethanol, centrifuged at 300× *g* for 15 s and supernatant discarded. The pellet was re-suspended in 500 µL of Acetone, centrifuged at 300× *g* for 15 s, supernatant discarded. The opened tube was placed in a heat block at 50 °C for 5 min, to dry the silica pellet. The pellet was re-suspended in 150 µL PCR grade water, and heated at 56 °C for 5 min, centrifuged at maximum speed for 20 min. Finally, 100 µL of supernatant containing RNA was transferred to sterile closed 0.5 mL tube, stored at −20 °C until further analysis.

#### 4.4.2. mPCR Detection of Sapoviruses from Stools

The RIDA^®^GENE Sapovirus, real-time RT-PCR kit (R-Biopharm AG, Darmstadt, Germany) for the direct detection of SV was used. This is a designed multiplex real-time RT-PCR for direct qualitative detection of human SV (Genogroup I, II, IV, and V), targeting the ORF1 region with fluorogenic target-specific hydrolysis probes. Reagents for the assay are provided with the kit including the internal control RNA which monitor PCR inhibition and reagent integrity. Prior PCR reaction, a 0.1 mL sterile tube with a total volume of 25 µL: containing 5 µL of RNA and 20 µL of Master Mix (19.3 µL of reaction mix, 0.7 µL of enzyme mix), 1 µL of internal control RNA was added to the negative and positive controls. The real-time PCR program was performed on a Corbett Research Rotor Gene 6000 with the following cycling conditions: Reverse transcription for 10 min at 58 °C; initial denaturation for 1 min at 95 °C followed by 45 cycles of 95 °C for 15 s and 60 °C for 60 s with continuous fluorescence reading, as per the manufacturer. To minimize the risk of sample contamination and amplicon carry-over, separate rooms were used for the pre- and post-amplification steps.

#### 4.4.3. Sapovirus Genogrouping and Sequencing

Positive samples for SV were further analyzed using One-Step Ahead RT-PCR kit (QIAGEN Co., Hilden, Germany) using previously published primers [6] to determine specific SV genogroups. The One Step Ahead RT-PCR utilizes a pair of specific oligonucleotide primers, namely: SV-G1-R/SaV124F to amplify GI capsid fragment, SV-G2-R/SaV124F to amplify GII capsid fragment, SV-G4-R/SaV124F to amplify GIV capsid fragment, and SV-G5-R/1245Rfwd to amplify the GV capsid fragment. Three randomly selected samples (SV-G1-R/SaV124F amplicons: Z01 [C_t_ = 23.42], Z22 [C_t_ = 22.21] and Z31 [C_t_ = 32.85]) were subjected to sequencing analysis. The PCR products of the amplified fragments were directly purified with a master mix of ExoSAP (Nucleics Pty Ltd., Woollahra, Australia). Using the same specific primers, Sanger sequencing was performed on the ABI 3500XL Genetic Analyzer POP7TM (Thermo-Scientific Inc., Waltham, MA, USA). The nucleotide of the successful sequences were compared with those of the reference strains available in the NCBI GenBank using BLAST tool [44]. Since Sapovirus is closely related to Norovirus, Noronet typing tool was used to determine the SV genotypes [25].

### 4.5. Statistical Analyses

Bayesian linear regression (ANOVA analysis) and Descriptive (Pearson χ^2^, 2-sided analysis) methods were performed for data analysis using IBM SPSS 26 software (IBM, Sandton, South Africa). Tests were used to determine statistical significance (*p* < 0.05).

## 5. Conclusions

The presence of SV in developing settings of the Vhembe region is evident. This is the first cross-sectional study to report on defined human SV strains in rural communities from South Africa. Outpatients in rural settings are potentially exposed to possible risk of the burden of diarrhoeal disease triggered by SVs among other pathogens and several factors including water, sanitation and hygiene practices. However, scientific data from Africa to report on enteric viruses as diarrhoeal causative agents are scant. Further investigation on the analysis and surveillance of human SV strains in rural settings (community or household level) is essential to assess burden of diseases.

## Figures and Tables

**Figure 1 viruses-13-00393-f001:**
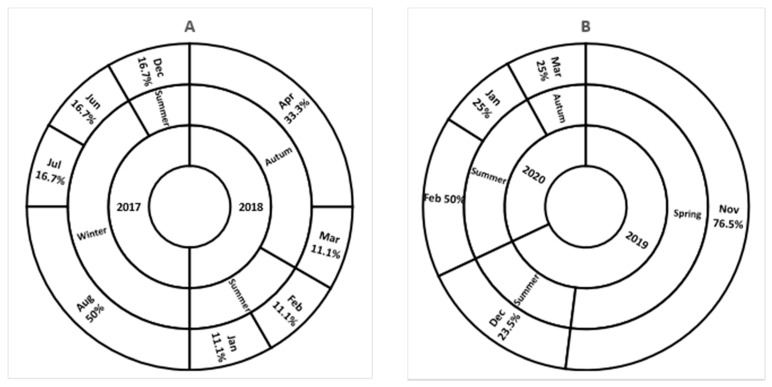
Seasonal distribution of detected SV from 2017 to 2018 (**A**), and from 2019 to 2020 (**B**). Inner circle present a year of sample collection; middle circle present seasons of the year; and the outer circle present the SV detection rate per month.

**Figure 2 viruses-13-00393-f002:**
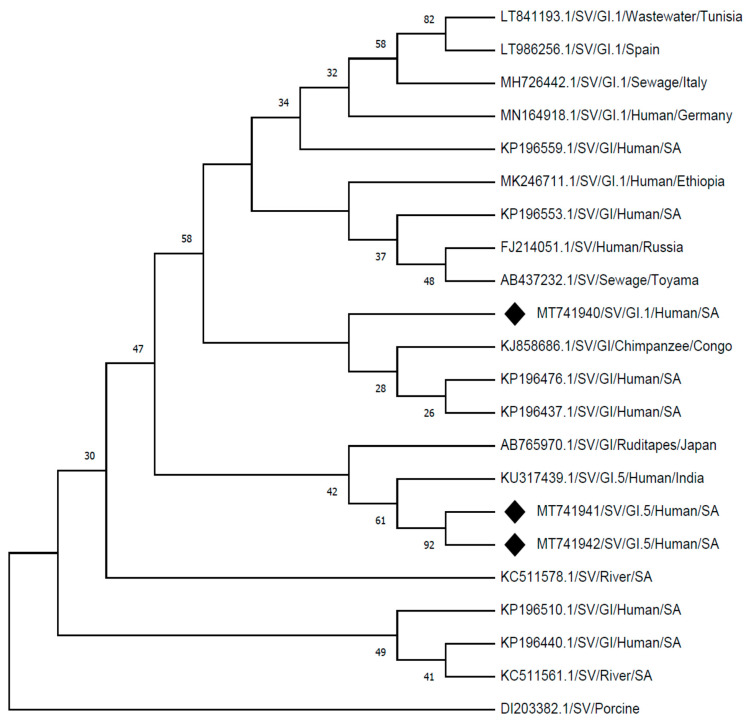
Phylogenetic analysis of the partial nucleotide sequences of sapovirus detected (MT741940, MT741941, and MT741942) in Vhembe district (South Africa), and reference strains of human sapovirus were selected from GeneBank database. Phylogenetic tree was deduced by the Neighbour-Joining method using MEGA X [26], based on a 360 base-pair fragment of the capsid (VP1) region showing the relationships within SV-G1 strains. The porcine SV (DI203382.1) was selected as an outgroup strain.

**Table 1 viruses-13-00393-t001:** Children presenting with diarrhoea and non-diarrhoea enrolled in the study.

Clinical Samples Collected	Outpatients from Clinics					Inpatients from Hospitals					Total Overall
	*n*	Age (Months)		Gender		*n*	Age (Months)		Gender		*n*
				M	F				M	F	
Children with diarrhoea (symptomatic)	137 (71%)	0–12	64(46.7%)	31	34	91 (100%)	0–12	46(50.5%)	28	18	228
		13–24	42(30.7%)	23	19		13–24	29(31.9%)	12	17	
		25–36	14(10.2%)	7	7		25–36	12(13.2%)	7	5	
		37–48	12(8.8%)	3	9		37–48	–	–	–	
		49–60	2(1.5%)	0	2		49–60	–	–	–	
		Unknown	3(2.2%)	1	2		Unknown	4(4.4%)	2	2	
Children without diarrhoea (asymptomatic)	56 (29%)	0–12	38(67.9%)	15	23	0 (0%)	0–12	–	N/A	N/A	56
		13–24	13(23.2%)	5	8		13–24	–			
		25–36	4(7.1%)	2	2		25–36	–			
		37–48	–	–	–		37–48	–			
		49–60	–	–	–		49–60	–			
		Unknown	1(1.8%)	0	1		Unknown	–			
Total	*n* = 193 (100%)					*n* = 91 (100%)					284

**Table 2 viruses-13-00393-t002:** Clinical features of study participant children under 5 years of age.

Case Patients (*n* = 228)			Controls (*n* = 56)	
Parameters	SV Positives (%) *n* = 38 (16.7%)	SV Negatives (%) *n* = 190 (83.3%)	SV Positives (%) *n* = 2 (3.6%)	SV Negatives (%) *n* = 54 (96.8%)
**Symptoms**			None	
Diarrhoea only	22 (57.9%)	68 (35.8%)		
Diarrhoea with other symptoms	16 (42.1%)	121 (63.7%)		
Unknown	–	1 (0.5%)		
**Other symptoms**			N/A	
Vomiting	11 (29.7%)	91 (47.6%)		
Fever	8 (21.6%)	60 (31.4%)		
Abdominal pain	4 (10.8%)	27 (14.1%)		
Dehydration	3 (8.1%)	24 (12.6%)		
**Interval ***			N/A	
≤3 days	22 (57.9%)	122 (64.2%)		
≥3 days	16 (42.1%)	67 (35.3%)		
Not defined	–	1 (0.5%)		

* Between the onset of diarrhoea and collection of sample. SV = Sapovirus.

**Table 3 viruses-13-00393-t003:** Symptoms shown by children in Clinics versus Hospital settings.

Case Patients (*n* = 228)		
Parameters	Clinics (*n* = 137) (Positives/No. of Cases (%))	Hospitals (*n* = 91) (Positives/No. of Cases (%))
Diarrhoea only	17/69 (24.6%)	5/21 (23.8%)
Diarrhoea with other symptoms	9/68 (13.2%)	7/69 (10.1%)
Unknown	None	1 (Neg)
**Other symptoms**		
Vomiting	7/45 (15.6%)	4/55 (7.3%)
Fever	4/22 (18.2%)	2/32 (6.3%)
Abdominal pain	2/13 (15.4%)	4/28 (14.3%)
Dehydration	0	3/27 (11.1%)

**Table 4 viruses-13-00393-t004:** Household settings of the participants and distribution of sapovirus (SV) positive cases.

Household Settings	SV Positives v/s Enrolled Cases (%)	Patients Age Group (Month) and SV Positive Cases					
		0–12 Months	13–24 Months	25–36 Months	37–48 Months	49–60 Months	Unknown
		19 pos	14 pos	4 pos	3 pos	0	0
**Latrine**							
Used	21/187 (11.2%)	9/96	7/53	2/24	3/10	0/2	0/2
Not used	19/95 (20%)	10/51	7/32	2/7	0/1	0	0/4
Unknown	0/2 (0%)	0	0	0/2	0	0	0
**Water sources**							
Tap	31/244 (12.7%)	14/125	11/72	4/30	2/10	0/1	0/6
Borehole	5/26 (19.2%)	3/16	1/6	0/2	1/1	0/1	0
River	2/2 (100%)	1/1	1/1	0	0	0	0
Spring	2/9 (22.2%)	1/5	1/4	0	0	0	0
Unknown	0/3 (0%)	0	0/3	0	0	0	0
**Breastfeeding**							
Yes	25/185 (13.5%)	16/119	6/45	2/9	1/7	0/1	0/4
No	14/92 (15.2%)	3/28	8/37	2/21	1/3	0/1	0/2
Unknown	1/7 (14.3%)	0/1	0/3	0/2	1/1	0	0
**Livestock**							
Present	9/88 (10.2%)	4/39	3/32	1/10	1/3	0	0/4
Absent	31/196 (15.8%)	15/106	11/56	3/22	2/8	0/2	0/2
Unknown	0/1 (0%)	0/1	0	0	0	0	0

SV = Sapovirus.

## Data Availability

Not applicable.

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
