# Peer review of "Prevalence and Genetic Characterisation of Human Sapovirus from Children with Diarrhoea in the Rural Areas of Vhembe District, South Africa, 2017–2020"

_viruses, 2021, doi:10.3390/v13030393_

Round 1
Reviewer 1 Report
Only a small amount of the genetic analysis has been accomplished as the authors stated in Discussion section.
Were 16 of the 40 samples that were able to amplify genogroup-specific RT-PCR at low Ct with the Sapovirus Real-Time (RT)-PCR kit? Such a summary is important for the reader.
Only 3 of the 16 samples that yielded PCR products could be sequenced, which is too low a success rate. Was there a single PCR product amplified in each primer set? It is recommended that you use purification methods other than ExoSAP or further cloning the purified PCR product at least confirm that these PCR amplicons are indeed the sequences of the virus of interest.
Table 2 and 3 can be put together. Table 4 is not necessary.
Figures 1 and 3 should be summarized in a Table.
For Figure 2, it would be easier to see the months on the horizontal axis.
Figure 4 requires the addition of genotype and other information.
There is no explanation for DI203382 in outgroup.
Author Response
Reviewer 1
Comments and Suggestions for Authors: Only a small amount of the genetic analysis has been accomplished as the authors stated in Discussion section.
Were 16 of the 40 samples that were able to amplify genogroup-specific RT-PCR at low Ct with the Sapovirus Real-Time (RT)-PCR kit? Such a summary is important for the reader.
Only 3 of the 16 samples that yielded PCR products could be sequenced, which is too low a success rate. Was there a single PCR product amplified in each primer set? It is recommended that you use purification methods other than ExoSAP or further cloning the purified PCR product at least confirm that these PCR amplicons are indeed the sequences of the virus of interest.
Table 2 and 3 can be put together. Table 4 is not necessary.
Figures 1 and 3 should be summarized in a Table.
For Figure 2, it would be easier to see the months on the horizontal axis.
Figure 4 requires the addition of genotype and other information.
There is no explanation for DI203382 in outgroup.
Author’s Response
We received comments from the reviewer which were relevant and critical to our article, and we addressed all issues suggested. As follows:
Reviewer 1
Comment:
Only a small amount of the genetic analysis has been accomplished as the authors stated in Discussion section.
Response: The authors agree with the reviewer. However, this study focused on the prevalence (number of sapovirus positives detected by real-time PCR) and the genogrouping of human sapovirus using commercial RT-PCR as stated in the abstract. The genetic analysis to determine the genotypes was randomly performed on some samples to confirm the genogrouping results by conventional RT-PCR results. However, the genotyping of all sapovirus positives which is from sequencing results will probably be in the future studies performed.
Line 206 (now line 226) in the discussion section was removed. The statement now reads as: “In this study, a high rate of SV detection in outpatient children…”
Comment:
Were 16 of the 40 samples that were able to amplify genogroup-specific RT-PCR at low Ct with the Sapovirus Real-Time (RT)-PCR kit? Such a summary is important for the reader.
Response: We agree, some of the samples with low Ct of sapovirus were amplified.
In this study the Ct range from RT-PCR was 22.21 – 38.97. 9 of 16 samples amplified had low RNA concentration (Ct > 34.12). Therefore, information on Ct value was added in results section (line 77,78).
Supporting information was added in lines 137 -142, and now reads as follows:
“With note, 9 of these samples had a low RNA concentration (Ct > 34.12). SV-GI (13/16: 81.3%) was predominately detected followed by SV-GII (2/16: 12.5%) and SV-GIV (1/16: 6.2%). Furthermore, 3 randomly selected samples (SV-G1-R/SaV124F amplicons of samples number Z01 [Ct = 23.42], Z22 [Ct = 22.21] and Z31 [Ct = 32.85]; from different clinics) were subjected to sequencing analysis to determine the SV genotypes.”
Comment:
Only 3 of the 16 samples that yielded PCR products could be sequenced, which is too low a success rate. Was there a single PCR product amplified in each primer set? It is recommended that you use purification methods other than ExoSAP or further cloning the purified PCR product at least confirm that these PCR amplicons are indeed the sequences of the virus of interest.
Response:
With note, the genotyping (sequencing) for all positive samples is in progress (The current 3 were randomly selected). The word “only” in line 117 (now line 139) was removed, and statement edited for clarity.
Each genogroup was detected by its specific pair of primers as stated in method section (subheading 4.4.3).
Recommendation for purification methods or cloning will be considered for further investigations.
Comment:
Table 2 and 3 can be put together. Table 4 is not necessary.
Response: Table 2 highlight data on clinical manifestation of all participants in this study, whereas Table 3 shows the differences on symptoms manifestation among the case patients, in different settings. Tables 2 and 3 were referred to in the discussion section, change of these tables will negatively affect the discussion section, therefore, we appeal to keep the tables separately.
We removed Table 4.
Comment:
Figures 1 and 3 should be summarized in a Table.
Response: Table 4 was added to summarize Figures 1 and 3 (Figures 1 and 3 were removed).
Results section: information supporting Table 4, line 107-117 was edited.
Comment:
For Figure 2, it would be easier to see the months on the horizontal axis.
Response: this is a technical challenge which could not be addressed because of the Microsoft Windows used, the option for rotating words is disabled.
Figure 2 is now Figure 1
Comment:
Figure 4 requires the addition of genotype and other information.
Response: We are not sure if the reviewer is suggesting addition of reference genotypes or genotypes detected from this study. Although this study was mainly focused on prevalence and genogrouping analysis, the phylogenetic tree was constructed to compare relatedness of the 3 randomly selected sapovirus strains with the strains detected from various sources/host and regions, motivated by the GenBank BLAST outcomes.
Comment:
There is no explanation for DI203382 in outgroup.
Response: DI203382 is a porcine Sapovirus strain used as an outgroup to better comparison for the selected SV strains.
Sentence added on the Figure 2 (previously Figure 4) heading
Line 153 - 156 was added to support.
Reviewer 2 Report
The study conducted on “Prevalence and genetic characterization of human sapovirus from children with diarrhoea in the rural areas of Vhembe district, South Africa, 2017-2020, by Magwalivha et al., is technically sound and interesting, but I have some minor comments as below.
Abstract
Abstract section is appropriate.
In line 12 Add space before and after “=”, revise throughout the manuscript.
Introduction
In my opinion, introduction is too short; author may add some physiopathological condition related to Sapovirus gastroenteritis. Why it is essential to detect Sapovirus from clinical samples? Are there any clinical significances of detecting this virus instead of Rotavirus and Norovirus (considered as major causes of gastroenteritis less than 5 years)?
Methods
Over all, method section is good but my concern is, did authors calculate the sample size?? Since sampling was done for 4 years (2017-2020), while the number of cases was too low compare to other studies conducted in South Africa.
In line 259 Nucleic acid extractions, authors haven’t mentioned the name, company, and city of the extraction kit. So, it would be nice to mention all details including volume of sample taken for nucleic acid extraction.
Similarly, for mPCR, it would be more effective if authors kindly add more information including volume taken for mPCR reaction, including PCR mixture, primers, and samples etc.
Results
Authors are requested to follow standard guideline for the result section. Please provide the sub-title for each section of results and please revise the table (standard format).
Discussion
Line 191 Replace word, “treatment” in place of “treatmet”
Line 218,219 There was statistical association between sources of water and positive case. Please define more for what purpose those water sources used in the study area. Whether it was used for drinking or other domestic purpose, please discuss more.
Conclusion
In line 297, 298 in my opinion better not to write enteric viruses, as long as in this study authors done analysis for only one virus.
Author Response
Reviewer 2
Comments and Suggestions for Authors: The study conducted on “Prevalence and genetic characterization of human sapovirus from children with diarrhoea in the rural areas of Vhembe district, South Africa, 2017-2020, by Magwalivha et al., is technically sound and interesting, but I have some minor comments as below.
Abstract
Abstract section is appropriate.
In line 12 Add space before and after “=”, revise throughout the manuscript.
Introduction
In my opinion, introduction is too short; author may add some physiopathological condition related to Sapovirus gastroenteritis. Why it is essential to detect Sapovirus from clinical samples? Are there any clinical significances of detecting this virus instead of Rotavirus and Norovirus (considered as major causes of gastroenteritis less than 5 years)?
Methods
Over all, method section is good but my concern is, did authors calculate the sample size?? Since sampling was done for 4 years (2017-2020), while the number of cases was too low compare to other studies conducted in South Africa.
In line 259 Nucleic acid extractions, authors haven’t mentioned the name, company, and city of the extraction kit. So, it would be nice to mention all details including volume of sample taken for nucleic acid extraction.
Similarly, for mPCR, it would be more effective if authors kindly add more information including volume taken for mPCR reaction, including PCR mixture, primers, and samples etc.
Results
Authors are requested to follow standard guideline for the result section. Please provide the sub-title for each section of results and please revise the table (standard format).
Discussion
Line 191 Replace word, “treatment” in place of “treatmet”
Line 218,219 There was statistical association between sources of water and positive case. Please define more for what purpose those water sources used in the study area. Whether it was used for drinking or other domestic purpose, please discuss more.
Conclusion
In line 297, 298 in my opinion better not to write enteric viruses, as long as in this study authors done analysis for only one virus.
Author’s Response
We received comments from the reviewer which were relevant and critical to our article, and we addressed all issues suggested. As follows:
Abstract
Abstract section is appropriate.
In line 12 Add space before and after “=”, revise throughout the manuscript.
Response: addition of spaces were done throughout the manuscript
Introduction
- In my opinion, introduction is too short; author may add some physiopathological condition related to Sapovirus gastroenteritis. Why it is essential to detect Sapovirus from clinical samples? Are there any clinical significances of detecting this virus instead of Rotavirus and Norovirus (considered as major causes of gastroenteritis less than 5 years)?
Response: the authors acknowledge the reviewer’s comment. Information to support the importance of SV detection was added.
Added information, line 39-46:
“Sapovirus has been associated with persistence vomiting suggested to possibly cause gastroenteritis in humans [7]. The increase of acute gastroenteritis associated with SVs has been reported, and recognized as a major public health problem particularly in developing countries [8,9]. It is documented that after the successful deployment of the Rotavirus vaccine, human SVs have emerged as the second most commonly etiological virus behind Norovirus in children with acute diarrhea [9]. In addition, a longitudinal study by MAL-ED reported SV as a notable second highest attributable incidence of diarrhea within the enrolled rural community in South Africa [10].”
Methods
- Over all, method section is good but my concern is, did authors calculate the sample size?? Since sampling was done for 4 years (2017-2020), while the number of cases was too low compare to other studies conducted in South Africa.
Response: we agree to the reviewer’s comment. Sample size was not calculated, number of samples collected were dependent on the willingness of participant to give sample during the researcher’s visit. The inconsistent collection of samples was also due to the Covid pandemic (National lockdown regulations).
- In line 259 Nucleic acid extractions, authors haven’t mentioned the name, company, and city of the extraction kit. So, it would be nice to mention all details including volume of sample taken for nucleic acid extraction.
Response:
Statement in line 259 (now 282) amended as:
“Prior to the nucleic acid extraction by Boom extraction reagents (Severn Biotech, UK)”
Details on extraction method was added in line 286-300:
“Briefly: A 500 µl of 10% suspension stool was centrifuged for 15 seconds (sec) at 12,000 rpm. 900 µl of L6 buffer was added to the supernatant in a sterile 1.5 ml tube, mixed by vortex for 1 minute (min), 20 µl of internal control was added, centrifuged for 15 sec @ 12,000 rpm. Into a sterile 1.5 ml tube, 100 µl of Silica beads was added to the transferred supernatant, mixed by vortex for 15 sec and shaken softly for 15 min. Tube was centrifuged at 2,000 rpm for 15 sec, and supernatant discarded. The pellet was re-suspended in 500 µl of L2 buffer, centrifuged at 2,000 rpm for 15 sec and supernatant discarded. The pellet was re-suspended in 500 µl of 70% Ethanol, centrifuged at 2,000 rpm for 15 sec and supernatant discarded. The pellet was re-suspended in 500 µl of Acetone, centrifuged at 2,000 rpm for 15 sec, supernatant discarded. The opened tube was placed in a heat block at 50 °C for 5 min, to dry the silica pellet. The pellet was re-suspended in 150 µl PCR grade water, and heated at 56 °C for 5 min, centrifuged at maximum speed for 20 min. 100 µl of supernatant containing RNA was transferred to sterile closed 0.5 ml tube, stored at -20 °C until further analysis.”
- Similarly, for mPCR, it would be more effective if authors kindly add more information including volume taken for mPCR reaction, including PCR mixture, primers, and samples etc.
Response:
Details on mPCR reaction volume and reagents was added, in line 307-310:
“Prior PCR reaction, a 0.1 ml sterile tube with a total volume of 25 µl: containing 5 µl of RNA and 20 µl of Master Mix (19.3 µl of reaction mix, 0.7 µl of enzyme mix), 1 µl of internal control RNA was added to the negative and positive controls.”
Results
- Authors are requested to follow standard guideline for the result section. Please provide the sub-title for each section of results and please revise the table (standard format).
Response: The results section have been amended as per the reviewer’s comment.
Discussion
- Line 191 Replace word, “treatment” in place of “treatmet”
Response: word corrected
Line 191, is now line 208
- Line 218,219 There was statistical association between sources of water and positive case. Please define more for what purpose those water sources used in the study area. Whether it was used for drinking or other domestic purpose, please discuss more.
Response:
Line 218-220, now Line 238-241, was amended and now reads as:
“municipal tap water as source of water (Table 4). Positive cases associated with water sources (variables: municipal tap water, borehole, river and spring) accessible by people for drinking, food preparation, bathing and other daily household tasks was established with statistical significance.”
See discussion, line 241 -244, 252-255
Conclusion
- In line 297, 298 in my opinion better not to write enteric viruses, as long as in this study authors done analysis for only one virus.
Response: We agree with the reviewer, Line 298 (now line 342), was amended and now reads as:
“the burden of diarrhoeal disease triggered by SV among other pathogens”
Round 2
Reviewer 1 Report
The explanation in the text is now clearer with the statement that “description of Ct values” and “sequencing of randomly selected strains”in the revised manuscript.
However, the description of the figures has not yet been improved.
In particular, I don't agree with posting the current phylogenetic analysis result.
New Fig1 (previous Fig2).
If you use this diagram, please set it to no color. This is unnecessary coloring.
New Fig.2 (previous Fig.4)
The current figure points out that the reader cannot understand which strain is of which genotype unless they checks from the accession number. This is a very taxing task for the reader!
If you don't want to include such information, then the sentence that “the detected strains were GI.1 and GI.5” is sufficient.
Author Response
Reviewer 1 (Round 2)
Comments and Suggestions for Authors
The explanation in the text is now clearer with the statement that “description of Ct values” and “sequencing of randomly selected strains” in the revised manuscript.
However, the description of the figures has not yet been improved.
In particular, I don't agree with posting the current phylogenetic analysis result.
New Fig1 (previous Fig2).
If you use this diagram, please set it to no color. This is unnecessary coloring.
New Fig.2 (previous Fig.4)
The current figure points out that the reader cannot understand which strain is of which genotype unless they checks from the accession number. This is a very taxing task for the reader!
If you don't want to include such information, then the sentence that “the detected strains were GI.1 and GI.5” is sufficient.
Author’s response:
We attempted to address issues suggested by the reviewer.
Comment:
New Fig1 (previous Fig2).
If you use this diagram, please set it to no color. This is unnecessary coloring.
Response:
Coloring was changed, as suggested by the reviewer.
Comment:
New Fig.2 (previous Fig.4).
The current figure points out that the reader cannot understand which strain is of which genotype unless they checks from the accession number. This is a very taxing task for the reader!
If you don't want to include such information, then the sentence that “the detected strains were GI.1 and GI.5” is sufficient.
Response:
Figure was edited so that the reader may be able to understand the types of strains used, without intensive checking.
Information on detected strains was stated in line 147,150-151.
Reviewer 2 Report
The current version of manuscript is acceptable for me. Still I would like to request authors to take care of figure section. Which is needed to be improved, the current version is difficult to read and understand.
Author Response
Reviewer 2 (Round 2)
Comments and Suggestions for Authors
The current version of manuscript is acceptable for me. Still I would like to request authors to take care of figure section. Which is needed to be improved, the current version is difficult to read and understand.
Author’s response:
We attempted to address issues suggested by the reviewer by also taking in account what the other reviewer requested from us on the Figures.
Comment:
Still I would like to request authors to take care of figure section. Which is needed to be improved, the current version is difficult to read and understand.
Response:
Background coloring of Figure 1 was changed, so that it can be clear to read.
Figure 2 was edited so that it become easier for the reader to understand it.